# Essential Oil Composition and Traditional Uses of *Salvia dentata*, a Poorly Known Medicinal Plant from Namaqualand, South Africa

**DOI:** 10.3390/molecules27196478

**Published:** 2022-10-01

**Authors:** Ryan D. Rattray, Lucky Mokwena, Marietjie A. Stander, Ben-Erik Van Wyk

**Affiliations:** 1Department of Botany and Plant Biotechnology, University of Johannesburg, P.O. Box 524, Auckland Park, Johannesburg 2006, South Africa; 2Central Analytical Facilities (CAF), J.C. Smuts Building Room 255, Faculty of Science, University of Stellenbosch, Private Bag X1, Matieland 7602, South Africa

**Keywords:** Lamiaceae, *Salvia*, essential oils, GC-MS, camphor, *Salvia africana*, *Salvia chamelaeagnea*, *Salvia dentata*, medicinal plants, South Africa

## Abstract

South Africa has a rich history of medicinal plant species and their documented uses as traditional medicines, and is also home to three well-known, blue-flowered sage species of ethnobotanical importance. The Namaqualand *bloublomsalie* (*Salvia dentata*) has so far remained unstudied and apparently overlooked. Our study is the first to report on the essential oil chemistry of this medicinally relevant species and provide a comparison with the other two (well-studied) closely related Cape *bloublomsalies* (*Salvia africana* and *S. chamelaeagnea*). The data, generated from three geographically isolated populations comprised of 13 individual plants of *S. dentata*, revealed diagnostically high levels of camphor (14.37%), α-pinene (11.43%), camphene (10.18%), 1,8-cineole (eucalyptol) (9.42%) and bornyl acetate (8.56%) which provide a distinct chemical profile from the other two species.

## 1. Introduction

The genus *Salvia* L. is the largest genus within the Lamiaceae (ca. 1010 spp. [1]) with various species used worldwide in traditional and folk medicines, treating ailments such as angiogenesis, bacterial and viral infections, inflammation and oxidative stress [2]. Although the genus was recognized by both Egyptian and the Greek civilizations, the name *Salvia* is derived from the Latin word “*salvēre*”, meaning “to feel healthy, to heal” given by the Roman Empire [3,4]. Many members of this genus are well known, such as *S. officinalis* L. (common sage), *S. rosmarinus* Spenn. (rosemary; hitherto better known as *Rosmarinus officinalis* L. [5]), *S. officinalis* subsp. *lavandulifolia* (Vahl) Gams (Spanish sage) and *S. fruticosa* Mill. (Greek sage) [6], largely for their therapeutic effects owed to their bioactive constituents [7]. These constituents include compounds such as 1,8-cineole (eucalyptol), α-thujone, camphor and camphene found in their essential oil (EO), as well as phenolic compounds such as caffeic-, rosmarinic- and salvianolic acid. In terms of flavonoids, 6-hydroxyflavones are characteristic of *Salvia* species and can be used as a chemosystematic/chemophenetic marker [8]. Being the largest genus within the Lamiaceae, the species have a wide geographical distribution, with the greatest concentration in Mexico and the Mediterranean region. In Asia, more than 40 species of *Salvia* are indigenous to China, with *S. miltiorrhiza* Bunge (red sage) being a popular herb in traditional Chinese medicine locally referred to as ‘*Danshen*’ or ‘*Tanshen*’ which is used to treat cardiovascular and renal ailments [7]. *Salvia hispanica* L., native to Mexico and Ecuador, is used both medicinally and as a source of protein where the seeds of this species are used as the notable superfood popularly known as ‘*chia*’ [9]. 

Southern Africa is home to 28 *Salvia* species, 14 of which are endemic to the region [10]. The Cape region of South Africa houses, amongst others, three closely related species of blue-flowered sages that are used in traditional medicines and are locally referred to as *bloublomsalies* (“blue flower sages”), and are *Salvia africana* L., *S. chamelaeagnea* Berg. and *S. dentata* Aiton (Figure 1A–C). The trichomes (hairs and glands) on the calyces are of diagnostic value used to distinguish between the three species [11]. *Salvia africana* and *S. chamelaeagnea* have been well studied, both in terms of their ethnobotanical records, pharmacological effects and chemistry [12,13]. The third species however, which is centered in Namaqualand (Figure 2)—is less well known and requires further investigation. Some traditional uses of *S. africana* and *S. chamelaeagnea* include the treatment of coughs, colds, influenza, chest troubles, headaches and fever [14,15]. *Salvia dentata* is used for the treatment of coughs, colds, influenza and stomach complaints [16,17]. Furthermore, an overlap exists between the traditional uses of these three medicinal *Salvia* species by various cultural groups of the Western and Northern Cape provinces, especially in the treatment of respiratory and gastrointestinal ailments. No publications could be found regarding the chemistry and essential oil composition of *S. dentata*. The aim of this paper is to present a review of all recorded ethnobotanical information about these three medicinally important species and to report for the first time the volatile components of *S. dentata* and compare the essential oil chemistry to that of the other two well-studied species.

## 2. Results and Discussion

### 2.1. Morphological Characteristics

Although the three species of Salvia appear to be morphologically similar, subtle differences exist by which the species can be distinguished from each other. The most useful and obvious diagnostic character is the pubescence of the calyx. The following characters and character states have been extracted from Codd [11] and can be used in identifying each species:

#### 2.1.1. *Salvia dentata*

A twiggy erect shrub ca. 2 m tall with greyish-tomentulose, gland-dotted stems (Figure 3A). Inflorescence of 2–9 spaced or crowded verticils, 2–6-flowered (Figure 3B,C). The calyx is somewhat funnel shaped, hispid and generally copiously red gland-dotted and occasionally hispid-villous (Figure 3D). The corolla is light blue or whitish to violet-blue to purple (Figure 3B–D). 

#### 2.1.2. *Salvia africana*

A shrub ca. 2 m tall which often branches at the base (Figure 4A) with several erect, usually sparingly branched stems of which are greyish-tomentulose to hispidulous and gland-dotted with occasional glandular hairs. Inflorescences are often dense or spaced below with 5–12 verticils which are 2–6-flowered (Figure 4B). The calyx is somewhat funnel-shaped, glandular-villous and purple tinged (Figure 4B). The corolla is light blue to bluish purple or pinkish with the lower lip usually with a paler blue margin and white to yellowish in the center (Figure 4B). 

#### 2.1.3. *Salvia chamelaeagnea*

A much-branched shrub 0.6–2 m tall (Figure 5A) with stems that are scabrid to pilose and gland-dotted. Inflorescences are large panicles, 100–300 mm long, with verticils two-flowered (Figure 5B). The calyx is reddish purple, glandular-hispid and gland-dotted. The corolla is blue or purplish blue often with white on the lower lip (Figure 5C,D).

### 2.2. Ethnobotanical Data

For the three *Salvia* species, the overall use categories with the highest number of use-records are respiratory system (32 records), gastrointestinal (16 records), analgesic (9 records), reproductive system (8 records) and topical applications (7 records; Figure 6). The ethnobotanical data show that *Salvia dentata* is mainly used for the treatment of respiratory ailments (colds, coughs and influenza), topical application, gastrointestinal (stomach complaints and diarrhoea) and unspecified medicinal uses (Table 1). Interestingly all three of the species are documented to be used as treatment for similar ailments (colds, coughs, influenza, stomach complaints and woman’s ailments) by different cultural groups across the Western and Northern Cape provinces. 

In South America *Salvia oppositiflora* Ruiz & Pav., *S. tubiflora* Sm., *S. dombeyi* Epling and *S. revoluta* Ruiz & Pav. are used by indigenous people primarily for the treatment of respiratory ailments. Secondary uses include analgesic, stomach concerns and topical uses [18]. Similarly, in the East Mediterranean, *S. fruticosa* Mill. [=*S. libanotica* Boiss. & Gaill.] is a popular medicinal plant that is widely used in the region. In Lebanon, Syria and Jordan a tea is made from the leaves which are commonly sold at the market and drunk for abdominal pain, headaches, stomach complaints and several other disorders. In Palestine it used for the treatment of stomach and cardiovascular issues. Furthermore, in Turkey, the plant is used to treat colds, coughs, influenza, and for gall bladder and kidney stones [19].

There seems to be a cultural overlap regarding the ethnobotanical uses across the globe for *Salvia* species in the treatment for certain conditions (i.e., respiratory, gastrointestinal, and topical). 

**Table 1 molecules-27-06478-t001:** Recorded ethnobotanical uses for the three medicinal, blue-flowered sages.

Species	Ethnobotanical Uses	Reference
*Salvia africana*	leaves: coughs, colds, chest troubles, uterine troubles, whooping cough; steam: infusion mixed with Epsom salts, lemon for abdominal troubles, diarrhoea, colic, flatulence, indigestion, veterinary (given to cows to expel placenta).	[20]
	leaves: coughs, colds, women’s ailments, abdominal troubles, fevers, measles,	[21]
	leaves: cough, colds, women’s ailments, chest ailments, fever, measles, stomach ailments, topical (bed sores, verrucose veins),	[22]
	leaves: coughs, colds, chest troubles, convulsions, stomach pain, flatulence, colic, woman’s ailments, diarrhoea	[14]
	leaves: coughs, colds, menstrual cramps, diarrhoea	[15]
	unspecified medicinal use	[23]
	leaves: tea, flavourant; flowers: nectar (snack)	[24]
	leaves: tea, flavourant; flowers: nectar (snack)	[25]
	leaves: burns, chest complaints, influenza, headache, fever, stomachache, stomach tonic, earache	[26]
*Salvia chamelaeagnea*	leaves: coughs, colds, bronchitis, diarrhoea, diaphoretic, female ailments, convulsions	[20]
	“also used medicinally”	[21]
	leaves: coughs, colds, chest troubles, convulsions, stomach pain, flatulence, colic, woman’s ailments, diarrhoea	[14]
	leaves: burns, chest complaints, influenza, headache, fever, stomachache, earache, dialarhoea	[15]
	leaves: colds, influenza, pain, inflammation, stroke, stomachache, topical (wash)	[27]
	unspecified medicinal use	[23]
	leaves: burns, chest complaints, influenza, headache, fever, stomachache, stomach tonic, earache	[26]
*Salvia dentata*	leaves and flowers: influenza	[28]
	leaves: influenza	[29]
*Salvia dentata* Cont.	leaves: leaf decoctions used for various ailments, often mixed with other herbs, general tonic, colds, backache, kidney diseases	[30]
	leaves: colds, influenza	[16]
	leaves: leaf decoctions used for coughs, colds, women’s ailments, diarrhoea	[31]
	leaves: colds, influenza, stomach complaints, unspecified medicinal use, measles; other: firewood	[17]
	unspecified medicinal use	[23]

### 2.3. Chemical Data

For comparative reasons, EOs were distilled from fresh material of *Salvia africana*, *S. chamelaeagnea* and *S. dentata* collected from several localities within the Northern and Western Cape provinces of South Africa (Figure 2; data points A–I). The mean percentage yield of EOs obtained for each group was 1.23% for *S. dentata*, 0.56% for *S. africana* and 0.54% for *S. chamelaeagnea*. The mean percentage yield obtained for *S. africana* was higher than results reported by other studies (0.17% [12], 0.13% [32] and 0.14% [33]), whereas the mean percent yield for *S. chamelaeagnea* were similar to those reported by Kamatou et al. [32] (0.39%) and Lim Ah Tock et al. [34] (0.43%). Higher EO yields are likely due to factors such as vegetative state, temperature, climate, geology and time of harvest [35,36]. Furthermore, the higher yield of *S. dentata* EO could be as a direct result of the species having a higher density of glandular trichomes on the leaf surface, which was observed for *S. africana* and *S. chamelaeagnea* by Kamatou et al. [32]. The essential oil obtained from *S. dentata*, *S. africana* and *S. chamelaeagnea* presented a pale-yellow color and all of the samples were aromatic. Further, *S. dentata* presented a highly aromatic odor distinct from the other two species.

Essential oils were analyzed by gas chromatography (GC) and gas chromatography-mass spectrometry (GC-MS). A total of 89 compounds were identified across three species (Table 2, Table 3 and Table 4) with *S. dentata* having the range 76.66–98.62% of compounds identified, *S. africana* with 81.45–94.29% of compounds identified and *S. chamelaeagnea* having 79.75–91.89% of compounds identified. Oxygenated monoterpenes were the dominant class of constituents in *S. dentata* (mean value of 35.21%), followed by monoterpene hydrocarbons, sesquiterpene hydrocarbons, oxygenated sesquiterpenes and hydrocarbons (mean values of 33.7%, 10.28%, 10.25% and 2.42% respectively). The *S. africana* oils were dominated by monoterpene hydrocarbons, oxygenated sesquiterpenes, oxygenated monoterpenes and sesquiterpene hydrocarbons (mean values of 36.02%, 24.92%, 16.08% and 11.58% respectively) and *S. chamelaeagnea* were dominated by oxygenated sesquiterpenes, oxygenated monoterpenes, sesquiterpene hydrocarbons and oxygenated monoterpenes (mean values of 30.78% 30.47%, 13.61% and 10.22% respectively). The percentage of grouped components for *S. africana* differed to a previous study by Kamatou et al. [32] where the authors reported 8.8% monoterpene hydrocarbons, 3.0% oxygenated monoterpenes, 7.5% sesquiterpene hydrocarbons and 58.7% oxygenated sesquiterpenes. Further, for *S. chamelaeagnea*, the same study reported 9.0% monoterpene hydrocarbons, 6.2% oxygenated monoterpenes, 16.6% sesquiterpene hydrocarbons and 47.9% oxygenated sesquiterpenes, ratios which are comparable to grouped compounds for *S. chamelaeagnea* EO identified in this study.

For *S. dentata*, the most abundant compounds (those with a peak area > 5%) were camphor (14.37%), α-pinene (11.43%), camphene (10.18%), 1,8-cineole (eucalyptol; 9.42%) and bornyl acetate (8.56%). 

In *S. africana* EO the trend of viridiflorol (19.74%) > β-caryophyllene (8.8%) > α-pinene (8.25%) > limonene (7.81%) > 1,8-cineole (eucalyptol) (6.02%) was observed. *Salvia chamelaeagnea* EO showed to be dominated by viridiflorol (24.57%) > limonene (10.58%) > β-caryophyllene (5.10%). Similar patterns in major compounds have been reported for *S. africana* by Kamatou et al. [33] and Van Vuuren et al. [37]. Furthermore, Lim Ah Tock et al. [34] observed viridiflorol (33.00%) and limonene (17.30%) being the major compounds for *S. chamelaeagnea*, where similarly Van Vuuren et al. [37] also noted viridiflorol (32.50%), limonene (14.10%) and 1,8-cineole (14.1%) as the major compounds in their study. The varying content of constituents observed in the extracted EOs among the three species are likely dependent on environmental factors such as altitude, climate, water availability and pedoclimatic conditions [35]. Further, factors such as genetics and phenological period have been shown to have an effect on the yield and content of volatile oils in aromatic plant species [35,38].

**Table 2 molecules-27-06478-t002:** Identified essential oil components (%) of *Salvia dentata*.

			*Salvia dentata*													
			Nieuwoudtville			Spoegrivier Road				Tankwa						
			Sd1	Sd2	Sd3	Sd4	Sd5	Sd6	Sd7	Sd8	Sd9	Sd10	Sd11	Sd12	Sd13	Mean
**Components/Yield (% dry weight)**	**RRI**	**ID**	0.99	0.71	1.38	1.25	1.14	1.30	1.20	1.56	1.76	0.95	1.70	1.06	1.01	1.23 ± 0.31
tricyclene	921	1, 2	0.33	0.00	0.00	0.61	0.45	0.54	0.00	0.64	0.00	0.78	0.72	0.50	0.66	0.4 ± 0.3
α-thujene	927	1, 2	0.00	0.00	0.00	0.13	0.16	0.13	0.13	0.17	0.00	0.00	0.00	0.00	0.12	0.06 ± 0.07
α-pinene	934	1, 2	**9.81**	**5.87**	**15.44**	**12.16**	**10.74**	**11.17**	**12.70**	**12.88**	**13.35**	**8.98**	**13.30**	**9.93**	**12.27**	**11.43 ± 2.42**
camphene	948	1, 2	**8.52**	**6.08**	**14.06**	**11.17**	**8.62**	**10.44**	**10.51**	**11.51**	**11.33**	**6.27**	**12.32**	**9.64**	**11.88**	**10.18 ± 2.32**
verbenene	963	1, 2	0.00	0.00	**5.41**	0.00	0.00	0.00	0.00	0.00	0.00	0.00	0.00	0.00	0.00	0.42 ± 1.5
sabinene	973	1, 2	0.00	0.00	0.06	0.05	0.07	0.06	0.04	0.00	0.07	0.12	0.05	0.06	0.05	0.05 ± 0.03
β-pinene	976	1, 2	2.19	1.39	2.58	2.50	2.18	2.37	2.17	2.64	3.38	3.14	3.50	3.02	2.74	2.6 ± 0.57
myrcene	990	1, 2	0.57	0.64	0.76	0.93	1.12	0.59	0.77	0.62	0.86	0.71	0.46	0.49	0.35	0.68 ± 0.21
α-phellandrene	1002	1, 2	0.00	0.27	0.16	0.37	0.27	0.26	0.00	0.31	0.49	0.04	0.06	0.36	0.07	0.2 ± 0.16
δ-3-carene	1009	1, 2	0.00	3.67	0.00	4.90	2.52	3.03	2.22	3.80	3.55	3.46	0.00	3.36	0.28	2.37 ± 1.72
α-terpinene	1015	1, 2	0.46	0.37	0.56	0.54	0.55	0.49	0.54	0.60	0.53	0.48	0.32	0.34	0.00	0.45 ± 0.16
*para*-cymene	1023	1, 2	0.53	0.98	0.70	0.75	0.30	0.77	0.67	0.99	0.49	1.19	0.51	0.85	0.39	0.7 ± 0.26
sylvestrene	1025	1, 2	0.00	0.00	0.00	0.45	0.00	0.00	0.00	0.00	0.00	0.00	0.00	0.00	0.00	t
limonene	1028	1, 2	2.12	2.90	0.00	3.24	2.38	2.55	2.29	2.73	2.76	2.54	1.91	2.28	1.65	2.26 ± 0.8
β-phellandrene	1029	1, 2	0.00	0.00	0.00	0.00	0.00	0.00	0.00	0.00	0.00	0.00	0.00	0.00	0.00	nd
1,8-cineole (eucalyptol)	1031	1, 2	**13.49**	3.59	**7.42**	**11.86**	**12.05**	**6.74**	**13.87**	**10.19**	**12.95**	**6.21**	**9.22**	**9.58**	**5.30**	**9.42 ± 3.35**
*cis*-β-ocimene	1037	1, 2	0.34	1.01	0.70	0.00	0.20	0.00	0.48	0.36	0.00	0.00	0.00	0.00	0.22	0.25 ± 0.32
*trans*-β-ocimene	1047	1, 2	0.45	1.43	0.90	0.00	0.30	0.00	0.55	0.52	0.00	0.00	0.49	0.00	0.22	0.37 ± 0.43
γ-terpinene	1059	1, 2	1.11	0.94	1.06	1.01	1.11	0.97	1.07	1.16	1.09	0.92	0.67	0.69	0.58	0.95 ± 0.19
*cis*-sabinene hydrate	1068	1, 2	0.00	0.00	0.00	0.28	0.00	0.00	0.07	0.08	0.23	0.18	0.08	0.03	0.00	0.07 ± 0.1
terpinolene	1088	1, 2	0.34	0.41	0.43	0.38	0.33	0.39	0.35	0.43	0.43	0.56	0.00	0.39	0.18	0.36 ± 0.14
*trans*-sabinene hydrate	1098	1, 2	0.00	0.00	0.11	0.00	0.19	0.25	0.21	0.11	0.00	0.35	0.00	0.00	0.32	0.12 ± 0.13
linalool	1099	1, 2	0.00	0.00	0.10	0.00	0.00	0.00	0.00	0.00	0.00	0.00	0.19	0.12	0.00	t
2-methylbutyl 2-methylbutyrate	1002	1, 2	0.00	0.00	0.00	0.00	0.10	0.00	0.00	0.08	0.00	0.00	0.00	0.00	0.00	t
solusterol	1103	1, 2	0.00	0.00	0.00	0.05	0.40	0.28	0.49	0.00	0.11	0.00	0.08	0.00	0.04	0.11 ± 0.17
*n*-amyl isovalerate	1107	1, 2	0.00	0.00	0.00	0.28	0.00	0.00	0.00	0.32	0.53	0.27	0.20	0.22	0.12	0.15 ± 0.17
α-thujone	1118	1, 2	0.26	0.00	0.03	0.21	0.00	0.09	0.06	0.00	0.21	0.37	0.21	0.26	0.16	0.14 ± 0.12
*cis*-para-menth-2-en-1-ol	1123	1, 2	0.00	0.00	0.00	0.00	0.00	0.00	0.00	0.04	0.00	0.10	0.06	0.06	0.00	t
α-campholene aldehyde	1128	1, 2	0.00	0.00	0.06	0.00	0.00	0.05	0.06	0.07	0.07	0.08	0.09	0.05	0.05	t
terpinen-1-ol	1142	1, 2	0.00	0.00	0.00	0.06	0.00	0.00	0.00	0.00	0.00	0.10	0.00	0.00	0.00	t
*trans*-verbenol	1142	1, 2	0.00	0.00	0.00	0.06	0.00	0.00	0.00	0.06	0.04	0.00	0.09	0.00	0.08	t
camphor	1148	1, 2	**17.70**	**14.57**	**8.93**	**19.16**	**15.15**	**18.07**	**18.46**	**18.66**	**18.06**	**7.36**	**12.31**	**9.87**	**8.45**	**14.37 ± 4.43**
camphene hydrate	1152	1, 2	0.00	0.00	0.00	0.06	0.00	0.06	0.00	0.07	0.06	0.00	0.00	0.00	0.00	t
exo-methyl-camphenilol	1153	1, 2	0.00	0.00	0.07	0.00	0.00	0.00	0.00	0.00	0.00	0.10	0.09	0.08	0.06	t
borneol	1170	1, 2	1.40	1.94	1.45	0.31	0.29	0.88	0.53	0.55	0.42	1.41	1.05	1.02	1.14	0.95 ± 0.51
terpinen-4-ol	1180	1, 2	0.00	1.00	0.85	0.56	0.54	0.59	0.69	0.70	0.76	1.04	0.72	0.85	0.60	0.69 ± 0.26
α-terpineol	1194	1, 2	0.00	0.00	0.09	0.18	0.08	0.05	0.21	0.25	0.08	0.14	0.08	0.11	0.09	t
myrtenol	1199	1, 2	0.00	0.00	0.00	0.00	0.00	0.00	0.00	0.00	0.00	0.00	0.05	0.00	0.00	nd
nopol	1206	1, 2	0.00	0.00	0.00	0.00	0.00	0.00	0.00	0.00	0.00	0.00	0.00	0.00	0.00	nd
*cis*-piperitol	1211	1, 2	0.00	0.00	0.00	0.00	0.00	0.00	0.00	0.00	0.00	0.00	0.00	0.00	0.00	nd
β-cyclocitral	1224	1, 2	0.00	0.00	0.00	0.00	0.00	0.00	0.00	0.00	0.00	0.00	0.00	0.00	0.00	nd
*cis*-3-hexenyl isovalerate	1235	1, 2	0.00	0.00	0.00	0.00	0.00	0.00	0.00	0.00	0.00	0.11	0.00	0.00	0.00	t
linalyl acetate	1255	1, 2	0.00	0.00	0.00	0.05	0.05	0.00	0.00	0.00	0.00	0.00	0.00	0.00	0.00	t
phellandral	1282	1, 2	0.00	0.00	0.00	0.00	0.00	0.00	0.00	0.00	0.00	0.00	0.00	0.00	0.00	nd
bornyl acetate	1284	1, 2	**11.90**	**21.68**	**6.34**	0.00	2.33	3.67	0.00	3.75	3.47	**8.47**	**14.39**	**15.41**	**19.88**	**8.56 ± 7.39**
α-fenchyl acetate	1298	1, 2	0.00	0.00	0.00	0.00	0.00	0.00	3.98	0.00	0.00	0.00	0.00	0.00	0.00	0.31 ± 1.1
myrtenyl acetate	1328	1, 2	0.00	0.00	0.00	0.00	0.00	0.00	0.00	0.00	0.00	0.00	0.00	0.00	0.00	nd
isoledene	1382	1, 2	0.00	0.00	0.00	0.15	0.24	0.04	0.15	0.19	0.00	0.20	0.11	0.10	0.23	0.11 ± 0.09
α-cubebene	1384	1, 2	0.00	0.00	0.05	0.18	0.20	0.00	0.00	0.00	0.39	0.15	0.00	0.00	0.06	0.08 ± 0.12
α-copaene	1385	1, 2	0.00	0.32	0.21	0.00	0.00	0.29	0.14	0.15	0.00	0.00	0.07	0.00	0.37	0.12 ± 0.14
α-ylangene	1387	1, 2	0.00	0.00	0.00	0.00	0.00	0.00	0.00	0.00	0.00	0.00	0.00	0.07	0.00	t
*cis*-jasmone	1403	1, 2	0.00	0.00	0.13	0.00	0.00	0.00	0.00	0.00	0.00	0.00	0.00	0.00	0.00	t
*cis*-β-caryophyllene	1417	1, 2	0.00	0.00	0.00	0.00	0.00	0.00	0.00	0.00	0.00	0.10	0.09	0.05	0.05	t
α-gurjunene	1420	1, 2	0.62	0.00	0.00	0.85	1.49	2.20	0.30	0.65	1.17	0.96	0.20	0.53	**0.58**	0.74 ± 0.62
β-caryophyllene	1431	1, 2	4.48	**9.00**	**6.13**	0.00	4.65	4.04	2.38	1.71	0.96	**5.88**	**8.44**	**6.41**	**5.34**	4.57 ± 2.74
clovene	1439	1, 2	0.00	0.00	0.00	0.00	0.00	0.00	0.00	0.00	0.00	0.00	0.00	0.00	0.00	nd
α-bergamotene	1443	1, 2	0.00	0.00	0.00	0.00	0.00	0.00	0.00	0.00	0.00	0.00	0.00	0.00	0.00	nd
calarene	1444	1, 2	0.00	0.00	0.18	0.21	0.26	0.17	0.54	0.00	0.12	0.11	0.15	0.06	0.23	0.16 ± 0.15
aromadendrene	1451	1, 2	1.93	0.00	0.06	2.76	0.00	2.54	2.50	2.11	0.00	0.00	1.89	0.00	4.29	1.39 ± 1.45
valencene I	1452	1, 2	0.00	0.00	0.00	0.00	3.62	0.00	0.00	0.00	0.00	0.00	0.00	1.13	0.00	0.37 ± 1.03
*trans*-β-farnesene	1458	1, 2	0.00	0.00	0.00	0.00	0.00	0.00	0.00	0.00	0.00	0.00	0.00	0.00	0.00	nd
α-humulene	1466	1, 2	0.54	1.01	0.00	0.00	0.46	0.41	0.00	0.18	0.13	0.00	1.00	0.86	0.78	0.41 ± 0.4
*allo*-aromadendrene	1473	1, 2	0.00	0.00	1.90	0.54	0.78	0.71	0.40	0.00	0.55	3.74	0.00	0.95	0.00	0.74 ± 1.05
γ-gurjunene	1484	1, 2	0.00	0.00	0.00	0.00	0.00	0.00	0.00	0.00	0.00	0.00	0.00	0.00	0.00	nd
α-amorphene	1486	1, 2	0.00	0.00	0.00	0.00	0.00	0.25	0.00	0.13	0.00	0.28	0.00	0.06	0.00	0.06 ± 0.1
β-ionone	1491	1, 2	0.00	0.00	0.00	0.00	0.00	0.00	0.00	0.00	0.00	0.00	0.00	0.00	0.00	nd
β-selinene	1495	1, 2	0.00	0.00	0.37	0.00	0.00	0.27	0.00	0.00	0.00	0.22	0.00	0.00	0.00	0.07 ± 0.13
germacrene-D	1499	1, 2	0.00	0.00	0.00	0.00	0.00	0.00	0.00	0.00	0.00	0.00	0.00	0.00	0.08	t
γ-himachalene	1501	1, 2	0.00	0.00	0.00	0.00	0.00	0.00	0.00	0.20	0.00	0.00	0.00	0.00	0.00	t
valencene	1506	1, 2	0.00	0.00	0.00	0.00	0.00	0.00	0.00	0.00	0.00	0.00	0.00	0.00	0.00	nd
ledene	1507	1, 2	0.73	0.00	0.93	1.11	1.88	1.24	0.87	0.00	1.00	1.01	0.60	0.52	1.16	0.85 ± 0.51
β-bisabolene	1513	1, 2	0.00	0.00	0.00	0.00	0.00	0.00	0.00	0.07	0.00	0.15	0.09	0.03	0.00	t
γ-cadinene	1524	1, 2	0.00	0.00	0.00	0.23	0.19	0.00	0.00	0.00	0.25	0.63	0.27	0.31	0.00	0.14 ± 0.19
β-cadinene	1532	1, 2	0.36	0.00	0.70	0.53	0.49	0.77	0.24	0.36	0.64	0.06	0.00	0.03	1.30	0.42 ± 0.38
nerolidol	1565	1, 2	0.00	0.00	2.89	1.95	1.35	1.20	1.48	1.54	1.58	2.75	0.76	1.50	1.48	1.42 ± 0.86
epi-globulol	1573	1, 2	0.00	0.00	0.00	0.00	0.00	0.00	0.00	0.00	0.00	0.00	0.00	0.00	0.00	nd
palustrol	1583	1, 2	0.64	0.00	0.00	2.22	2.87	**6.18**	0.00	1.80	2.94	2.96	0.00	2.15	0.00	1.67 ± 1.84
spathulenol	1590	1, 2	0.32	0.00	0.29	0.24	0.16	0.16	0.17	0.17	0.22	0.23	0.00	0.17	0.28	0.18 ± 0.1
caryophyllene oxide	1597	1, 2	0.00	0.39	1.69	0.00	0.00	0.00	0.00	0.48	0.00	1.81	0.12	0.00	0.00	0.35 ± 0.64
globulol	1598	1, 2	0.00	0.00	0.18	0.57	0.52	0.00	0.00	0.00	0.00	0.00	0.00	0.00	0.00	0.1 ± 0.2
viridiflorol	1607	1, 2	1.28	0.00	**8.14**	**10.82**	**13.12**	1.77	**15.33**	**12.44**	**7.10**	0.00	2.20	3.31	1.79	**5.95 ± 5.48**
γ-eudesmol	1647	1, 2	0.00	0.00	0.00	0.00	0.00	0.00	0.00	0.00	0.00	0.00	1.09	0.86	0.50	0.19 ± 0.38
1(5),6-guaiadiene	1662	1, 2	0.00	0.00	0.00	0.00	0.00	0.00	0.00	0.00	0.00	0.00	0.00	0.00	0.00	nd
β-eudesmol	1665	1, 2	0.00	0.00	0.00	0.00	0.00	0.00	0.00	0.00	0.00	0.00	1.08	0.78	0.52	0.18 ± 0.37
α-eudesmol	1669	1, 2	0.00	0.00	0.00	0.00	0.00	0.00	0.00	0.00	0.00	0.00	2.28	0.00	0.00	0.18 ± 0.63
τ-muurolol	1666	1, 2	0.00	0.00	0.00	0.00	0.00	0.00	0.00	0.00	0.00	0.00	0.00	0.00	0.00	nd
α-bisabolol	1692	1, 2	0.00	0.00	0.00	0.00	0.00	0.00	0.00	0.06	0.00	0.00	0.00	0.17	0.11	t
tetracosane	2300	1, 2	**12.24**	0.00	0.00	0.00	0.00	0.00	0.00	0.00	0.00	0.00	0.00	0.00	0.00	0.94 ± 3.4
heneicosane	2264	1, 2	0.00	**19.17**	0.00	0.00	0.00	0.00	0.00	0.00	0.00	0.00	0.00	0.00	0.00	1.47 ± 5.32
% Identification			94.66	98.62	92.07	94.91	94.70	86.63	97.56	96.46	92.33	76.66	93.65	89.53	87.11	91.91 ± 5.88
**Grouped components:**																
Monoterpene hydrocarbons			26.77	25.96	42.80	39.19	31.29	33.75	34.44	39.36	38.34	29.14	34.30	31.91	31.70	33.77 ± 5.08
Oxygenated monoterpenes			44.76	42.78	25.43	33.35	31.14	30.69	38.63	34.89	36.95	26.29	38.91	37.61	36.25	35.21 ± 5.77
Sesquiterpene hydrocarbons			8.64	10.32	10.53	6.57	14.25	12.88	7.51	5.74	5.21	13.48	12.91	11.07	14.48	10.28 ± 3.27
Oxygenated sesquiterpenes			2.24	0.39	13.31	15.79	18.02	9.31	16.98	16.48	11.84	7.75	7.53	8.94	4.69	10.25 ± 5.74
Hydrocarbons			12.24	19.17	0.00	0.00	0.00	0.00	0.00	0.00	0.00	0.00	0.00	0.00	0.00	2.42 ± 6.07

RRI: Relative retention indices calculated against n-alkanes, ID: identification method—1 = comparison of retention index; 2 = comparison of mass spectra with MS libraries, nd: not detected, t: trace, major compounds (% > 5) highlighted in **bold**.

**Table 3 molecules-27-06478-t003:** Identified essential oil components (%) of *Salvia africana*.

			*Salvia africana*						
			Cederberg			Citrusdal	Elandskloof	Theronsberg Pass	
			Sa1	Sa2	Sa3	Sa4	Sa5	Sa6	Mean
**Components/Yield (% dry weight)**	**RRI**	**ID**	0.36	0.33	0.28	1.40	0.34	0.62	0.56 ± 0.43
tricyclene	921	1, 2	0.00	0.00	0.08	0.00	0.00	0.30	0.06 ± 0.12
α-thujene	927	1, 2	0.32	0.56	0.19	0.15	0.00	0.47	0.28 ± 0.21
α-pinene	934	1, 2	**5.61**	**9.64**	**11.45**	**8.34**	4.95	**9.53**	**8.25 ± 2.52**
camphene	948	1, 2	0.17	0.11	1.70	3.20	0.20	5.30	1.78 ± 2.11
verbenene	963	1, 2	0.00	0.00	0.00	0.00	0.00	0.00	nd
sabinene	973	1, 2	0.08	0.09	0.05	0.00	0.09	0.13	0.07 ± 0.04
β-pinene	976	1, 2	1.86	2.78	2.81	2.20	3.26	3.03	2.66 ± 0.53
myrcene	990	1, 2	1.50	1.18	1.25	1.27	0.81	0.90	1.15 ± 0.26
α-phellandrene	1002	1, 2	0.10	0.00	0.03	0.69	1.08	1.68	0.60 ± 0.68
δ-3-carene	1009	1, 2	0.07	0.08	0.06	0.00	7.83	1.69	1.62 ± 3.11
α-terpinene	1015	1, 2	0.33	0.08	0.11	0.46	0.42	0.68	0.35 ± 0.23
para-cymene	1023	1, 2	3.24	0.56	2.03	0.24	0.08	1.66	1.30 ± 1.23
sylvestrene	1025	1, 2	0.00	0.00	0.00	0.00	1.21	0.00	0.20 ± 0.50
limonene	1028	1, 2	3.68	1.11	1.53	**21.96**	**18.60**	0.00	7.81 ± 9.79
β-phellandrene	1029	1, 2	0.00	0.00	0.00	0.00	0.00	0.00	nd
1,8-cineole (eucalyptol)	1031	1, 2	4.18	**5.65**	1.08	**17.39**	0.29	**7.55**	6.02 ± 6.2
cis-β-ocimene	1037	1, 2	0.00	4.59	**6.36**	0.34	0.11	0.20	1.93 ± 2.8
trans-β-ocimene	1047	1, 2	**6.78**	**8.53**	**11.45**	0.75	0.00	0.09	4.6 ± 4.97
γ-terpinene	1059	1, 2	3.60	0.41	0.95	0.75	1.01	3.32	1.67 ± 1.4
cis-sabinene hydrate	1068	1, 2	0.13	0.12	0.00	0.09	0.00	0.00	0.06 ± 0.06
terpinolene	1088	1, 2	0.16	0.18	0.37	0.18	0.33	0.33	0.26 ± 0.09
trans-sabinene hydrate	1098	1, 2	0.00	0.19	0.29	0.00	0.24	0.61	0.22 ± 0.23
linalool	1099	1, 2	0.34	0.00	0.00	0.24	0.37	0.00	0.16 ± 0.18
2-methylbutyl 2-methylbutyrate	1002	1, 2	0.00	0.00	0.00	0.00	0.00	0.00	nd
solusterol	1103	1, 2	0.00	0.00	0.00	0.00	0.00	0.00	nd
n-amyl isovalerate	1107	1, 2	0.00	0.00	0.00	0.00	0.00	0.08	t
α-thujone	1118	1, 2	0.00	0.00	0.00	0.00	0.00	0.13	t
cis-para-menth-2-en-1-ol	1123	1, 2	0.04	0.00	0.00	0.29	0.08	0.15	0.09 ± 0.11
α-campholene aldehyde	1128	1, 2	0.00	0.00	0.00	0.00	0.00	0.03	t
terpinene-1-ol	1142	1, 2	0.00	0.00	0.00	0.00	0.07	0.11	t
trans-verbenol	1142	1, 2	0.00	0.00	0.00	0.00	0.00	0.00	nd
camphor	1148	1, 2	0.15	0.08	2.46	**6.63**	0.27	**5.39**	2.5 ± 2.89
camphene hydrate	1152	1, 2	0.00	0.00	0.00	0.00	0.00	0.00	nd
exo-methyl-camphenilol	1153	1, 2	0.00	0.00	0.00	0.00	0.00	0.04	t
borneol	1170	1, 2	0.06	0.00	0.00	0.22	0.10	0.59	0.16 ± 0.22
terpinene-4-ol	1180	1, 2	0.40	0.29	0.51	0.47	0.95	0.77	0.56 ± 0.25
α-terpineol	1194	1, 2	0.00	0.00	0.00	0.17	0.12	0.19	0.08 ± 0.09
myrtenol	1199	1, 2	1.09	1.17	0.96	0.00	3.66	1.21	1.35 ± 1.22
nopol	1206	1, 2	0.00	0.00	0.00	0.20	0.00	0.00	t
cis-piperitol	1211	1, 2	0.00	0.00	0.00	0.00	0.07	0.07	t
β-cyclocitral	1224	1, 2	0.00	0.00	0.07	0.07	0.04	0.03	t
cis-3-hexenyl isovalerate	1235	1, 2	0.00	0.00	0.00	0.00	0.00	0.00	nd
linalyl acetate	1255	1, 2	0.00	0.00	0.00	0.04	0.00	0.00	t
phellandral	1282	1, 2	0.00	0.00	0.00	0.00	0.04	0.00	t
bornyl acetate	1284	1, 2	0.12	0.09	**6.14**	0.00	0.00	**7.81**	2.36 ± 3.62
α-fenchyl acetate	1298	1, 2	0.00	0.00	0.00	0.00	0.00	0.00	nd
myrtenyl acetate	1328	1, 2	0.75	0.83	1.73	1.73	0.13	0.22	0.9 ± 0.7
isoledene	1382	1, 2	0.00	0.00	0.00	0.00	0.00	0.00	nd
α-cubebene	1384	1, 2	0.00	0.06	0.00	0.06	0.19	0.06	0.06 ± 0.07
α-copaene	1385	1, 2	0.00	0.00	0.00	0.13	0.05	0.30	0.08 ± 0.12
α-ylangene	1387	1, 2	0.00	0.00	0.00	0.00	0.00	0.00	nd
cis-jasmone	1403	1, 2	0.17	0.00	0.11	0.00	0.00	0.00	0.05 ± 0.08
cis-β-caryophyllene	1417	1, 2	0.00	0.06	0.03	0.00	0.00	0.03	t
α-gurjunene	1420	1, 2	0.00	0.00	0.67	0.00	0.09	0.41	0.2 ± 0.28
β-caryophyllene	1431	1, 2	**10.89**	**16.38**	**13.35**	0.72	**5.29**	**6.15**	**8.8 ± 5.78**
clovene	1439	1, 2	0.11	0.00	0.06	0.00	0.00	0.00	t
α-bergamotene	1443	1, 2	0.00	0.00	0.00	0.00	0.00	0.00	nd
calarene	1444	1, 2	0.03	0.04	0.00	0.00	0.06	0.06	t
aromadendrene	1451	1, 2	0.00	0.00	0.28	0.00	0.31	0.83	0.24 ± 0.33
valencene I	1452	1, 2	0.00	0.00	0.00	0.00	0.00	0.00	nd
trans-β-farnesene	1458	1, 2	0.00	0.00	0.00	0.62	0.00	0.00	0.1 ± 0.25
α-humulene	1466	1, 2	1.58	1.84	1.72	0.45	0.61	0.83	1.17 ± 0.61
allo-aromadendrene	1473	1, 2	0.02	1.64	0.32	0.03	1.64	0.00	0.61 ± 0.81
γ-gurjunene	1484	1, 2	0.00	0.00	0.00	0.00	0.00	0.00	nd
α-amorphene	1486	1, 2	0.00	0.00	0.00	0.00	0.35	0.00	0.06 ± 0.14
β-ionone	1491	1, 2	0.03	0.00	0.00	0.00	0.00	0.00	t
β-selinene	1495	1, 2	0.00	0.05	0.00	0.00	0.00	0.00	t
germacrene-D	1499	1, 2	0.00	0.00	0.00	0.00	0.00	0.00	nd
γ-himachalene	1501	1, 2	0.00	0.00	0.00	0.00	0.00	0.00	nd
valencene	1506	1, 2	0.56	0.00	0.00	0.00	0.11	0.00	0.11 ± 0.22
ledene	1507	1, 2	0.00	0.00	0.00	0.00	0.82	0.73	0.26 ± 0.4
β-bisabolene	1513	1, 2	0.03	0.19	0.00	0.00	0.06	0.08	0.06 ± 0.07
γ-cadinene	1524	1, 2	1.13	0.00	0.00	0.00	0.00	0.12	0.21 ± 0.45
β-cadinene	1532	1, 2	0.00	0.00	0.23	0.00	0.00	0.00	0.04 ± 0.09
nerolidol	1565	1, 2	1.37	1.52	0.84	0.00	0.82	0.99	0.92 ± 0.53
epi-globulol	1573	1, 2	0.00	0.10	0.00	0.00	0.33	0.00	0.07 ± 0.13
palustrol	1583	1, 2	0.00	0.00	4.24	0.00	0.00	1.32	0.93 ± 1.71
spathulenol	1590	1, 2	0.00	0.49	0.18	0.00	1.06	0.37	0.35 ± 0.4
caryophyllene oxide	1597	1, 2	3.89	2.90	2.98	0.00	0.00	0.99	1.79 ± 1.68
globulol	1598	1, 2	0.00	0.00	0.00	0.00	0.00	0.00	nd
viridiflorol	1607	1, 2	**16.47**	**29.00**	**11.69**	**23.12**	**24.98**	**13.19**	**19.74 ± 6.97**
γ-eudesmol	1647	1, 2	0.00	0.00	0.00	0.00	0.00	0.00	nd
1(5),6-guaiadiene	1662	1, 2	**13.97**	0.00	0.00	0.00	0.00	0.00	2.33 ± 5.7
β-eudesmol	1665	1, 2	0.00	0.69	0.21	0.00	0.00	0.42	0.22 ± 0.28
α-eudesmol	1669	1, 2	0.00	0.00	0.00	0.00	0.00	0.00	nd
τ-muurolol	1666	1, 2	0.20	0.00	0.00	0.00	0.34	0.37	0.15 ± 0.18
α-bisabolol	1692	1, 2	0.22	1.10	0.06	0.00	0.12	0.09	0.27 ± 0.42
tetracosane	2300	1, 2	0.00	0.00	0.00	0.00	0.00	0.00	nd
heneicosane	2264	1, 2	0.00	0.00	0.00	0.00	0.00	0.00	nd
% Identification			85.28	94.29	90.58	93.14	83.56	81.45	88.05 ± 5.34
**Grouped components:**									
Monoterpene hydrocarbons			27.52	29.91	40.38	40.55	39.97	29.30	34.61 ± 6.29
Oxygenated monoterpenes			7.22	8.41	13.23	27.49	6.36	24.91	14.6 ± 9.33
Sesquiterpene hydrocarbons			28.22	20.17	16.65	1.98	9.59	9.51	14.35 ± 9.28
Oxygenated sesquiterpenes			22.32	35.80	20.32	23.12	27.64	17.74	24.49 ± 6.44
Hydrocarbons			0.00	0.00	0.00	0.00	0.00	0.00	nd

RRI: Relative retention indices calculated against n-alkanes, ID: identification method—1 = comparison of retention index; 2 = comparison of mass spectra with MS libraries, nd: not detected, t: trace, major compounds (% > 5) highlighted in bold.

**Table 4 molecules-27-06478-t004:** Identified essential oil components (%) of *Salvia chamelaeagnea*.

			*Salvia chamelaeagnea*						
			Piketberg			Darling			
			Sc1	Sc2	Sc3	Sc4	Sc5	Sc6	Mean
**Components/Yield (% dry weight)**	**RRI**	**ID**	0.35	0.13	1.60	0.55	0.22	0.40	0.54 ± 0.54
tricyclene	921	1, 2	0.56	0.00	0.29	0.00	0.00	0.00	0.14 ± 0.24
α-thujene	927	1, 2	**7.16**	0.29	0.61	0.05	0.13	0.00	1.37 ± 2.84
α-pinene	934	1, 2	0.13	**7.16**	**7.07**	2.61	4.70	2.54	4.03 ± 2.79
camphene	948	1, 2	0.00	0.10	**6.40**	0.24	0.09	0.10	1.16 ± 2.57
verbenene	963	1, 2	0.00	0.00	0.00	0.00	0.00	0.00	nd
sabinene	973	1, 2	0.16	0.10	0.15	0.00	0.08	0.00	0.08 ± 0.07
β-pinene	976	1, 2	2.19	2.29	2.99	0.75	1.48	0.53	1.71 ± 0.96
myrcene	990	1, 2	2.80	1.62	1.18	0.95	1.95	0.79	1.55 ± 0.75
α-phellandrene	1002	1, 2	**13.76**	0.54	2.12	0.26	0.22	0.30	2.87 ± 5.39
δ-3-carene	1009	1, 2	0.07	0.11	0.09	0.00	0.17	0.00	0.07 ± 0.07
α-terpinene	1015	1, 2	2.50	0.43	0.00	0.18	0.35	0.17	0.61 ± 0.94
para-cymene	1023	1, 2	0.00	0.00	0.36	0.31	0.31	0.24	0.2 ± 0.16
sylvestrene	1025	1, 2	0.00	0.00	0.00	0.00	0.00	0.00	nd
limonene	1028	1, 2	0.00	**7.85**	0.00	**20.88**	**15.36**	**19.38**	**10.58 ± 9.36**
β-phellandrene	1029	1, 2	0.00	0.00	**12.68**	0.00	0.00	0.00	2.11 ± 5.18
1,8-cineole (eucalyptol)	1031	1, 2	3.29	1.22	1.80	**7.64**	**8.14**	4.95	4.51 ± 2.93
cis-β-ocimene	1037	1, 2	1.63	2.17	0.26	0.00	0.00	0.36	0.74 ± 0.93
trans-β-ocimene	1047	1, 2	2.88	1.09	0.15	1.98	**7.13**	0.93	2.36 ± 2.52
γ-terpinene	1059	1, 2	0.30	1.09	0.93	0.41	0.97	0.32	0.67 ± 0.36
cis-sabinene hydrate	1068	1, 2	0.06	0.00	0.00	0.00	0.00	0.00	t
terpinolene	1088	1, 2	0.41	0.35	0.38	0.00	0.27	0.11	0.25 ± 0.17
trans-sabinene hydrate	1098	1, 2	0.00	0.00	1.14	0.00	0.00	0.00	0.19 ± 0.47
linalool	1099	1, 2	0.23	0.15	0.00	0.32	0.20	0.14	0.17 ± 0.11
2-methylbutyl 2-methylbutyrate	1002	1, 2	0.00	0.00	0.00	0.00	0.00	0.00	nd
solusterol	1103	1, 2	0.00	0.00	0.00	0.00	0.00	0.00	nd
n-amyl isovalerate	1107	1, 2	0.00	0.00	0.00	0.00	0.00	0.00	nd
α-thujone	1118	1, 2	0.00	0.00	0.00	0.00	0.00	0.00	nd
cis-para-menth-2-en-1-ol	1123	1, 2	0.07	0.00	0.14	0.09	0.00	0.09	0.06 ± 0.06
α-campholene aldehyde	1128	1, 2	0.00	0.00	0.00	0.00	0.00	0.00	nd
terpinen-1-ol	1142	1, 2	0.00	0.00	0.09	0.07	0.00	0.00	t
trans-verbenol	1142	1, 2	0.06	0.00	0.00	0.00	0.00	0.00	t
camphor	1148	1, 2	0.06	0.11	**7.61**	0.14	0.06	0.08	1.34 ± 3.07
camphene hydrate	1152	1, 2	0.00	0.00	0.00	0.00	0.00	0.00	nd
exo-methyl-camphenilol	1153	1, 2	0.00	0.00	0.00	0.00	0.00	0.00	nd
borneol	1170	1, 2	0.00	0.00	**10.90**	0.00	0.00	0.00	1.82 ± 4.45
terpinen-4-ol	1180	1, 2	1.02	0.00	0.80	0.28	0.35	0.25	0.45 ± 0.38
α-terpineol	1194	1, 2	0.23	0.34	0.23	0.40	0.21	0.17	0.26 ± 0.09
myrtenol	1199	1, 2	0.47	2.03	4.91	0.00	0.00	0.00	1.23 ± 1.96
nopol	1206	1, 2	0.00	0.09	0.06	0.00	0.00	0.00	t
cis-piperitol	1211	1, 2	0.00	0.00	0.11	0.00	0.00	0.00	t
β-cyclocitral	1224	1, 2	0.11	0.09	0.06	0.00	0.00	0.00	t
cis-3-hexenyl isovalerate	1235	1, 2	0.00	0.00	0.00	0.00	0.00	0.00	nd
linalyl acetate	1255	1, 2	0.00	0.00	0.00	0.00	0.00	0.00	nd
phellandral	1282	1, 2	0.00	0.00	0.00	0.00	0.00	0.00	nd
bornyl acetate	1284	1, 2	0.00	0.00	0.29	0.00	0.00	0.00	0.05 ± 0.12
α-fenchyl acetate	1298	1, 2	0.00	0.00	0.00	0.00	0.00	0.00	nd
myrtenyl acetate	1328	1, 2	0.00	0.04	0.00	0.00	0.00	0.00	t
isoledene	1382	1, 2	0.08	0.05	0.06	0.00	0.30	0.21	0.12 ± 0.11
α-cubebene	1384	1, 2	0.00	0.00	0.00	0.00	0.19	0.10	0.05 ± 0.08
α-copaene	1385	1, 2	0.40	0.94	0.00	0.07	0.00	0.00	0.24 ± 0.38
α-ylangene	1387	1, 2	0.00	0.00	0.00	0.00	0.00	0.00	nd
cis-jasmone	1403	1, 2	0.36	0.00	0.06	0.00	0.00	0.00	0.07 ± 0.14
cis-β-caryophyllene	1417	1, 2	0.00	0.00	0.00	0.03	0.00	0.00	t
α-gurjunene	1420	1, 2	0.16	0.20	0.07	0.76	0.40	1.03	0.44 ± 0.38
β-caryophyllene	1431	1, 2	0.00	**14.76**	3.17	2.85	**5.32**	4.53	5.1 ± 5.07
clovene	1439	1, 2	0.00	0.00	0.00	0.00	0.00	0.00	nd
α-bergamotene	1443	1, 2	0.00	0.24	0.00	0.00	0.00	0.00	t
calarene	1444	1, 2	0.17	0.15	0.04	0.31	0.27	0.36	0.22 ± 0.12
aromadendrene	1451	1, 2	0.00	0.73	0.00	3.79	0.00	1.09	0.93 ± 1.47
valencene I	1452	1, 2	0.00	0.73	1.43	0.00	0.00	4.05	1.03 ± 1.58
trans-β-farnesene	1458	1, 2	0.00	0.00	0.00	0.00	0.00	0.00	nd
α-humulene	1466	1, 2	0.00	2.07	0.00	1.60	1.11	2.71	1.25 ± 1.1
allo-aromadendrene	1473	1, 2	**5.75**	0.00	0.00	0.89	**5.25**	0.00	1.98 ± 2.75
γ-gurjunene	1484	1, 2	0.00	0.00	0.00	0.00	0.15	0.21	0.06 ± 0.1
α-amorphene	1486	1, 2	0.00	0.00	0.00	0.00	0.00	1.12	0.19 ± 0.46
β-ionone	1491	1, 2	0.00	0.00	0.00	0.00	0.00	0.00	nd
β-selinene	1495	1, 2	0.00	0.00	0.00	0.00	0.00	0.00	nd
germacrene-D	1499	1, 2	0.00	0.00	0.00	0.00	0.00	0.00	nd
γ-himachalene	1501	1, 2	0.00	0.00	0.00	0.00	0.00	0.00	nd
valencene	1506	1, 2	0.00	0.00	0.00	0.00	0.00	0.00	nd
ledene	1507	1, 2	0.91	1.30	0.85	1.25	2.34	1.93	1.43 ± 0.59
β-bisabolene	1513	1, 2	0.00	0.31	0.00	0.00	0.00	0.00	0.05 ± 0.12
γ-cadinene	1524	1, 2	0.00	0.12	0.40	0.00	0.67	0.00	0.2 ± 0.28
β-cadinene	1532	1, 2	0.33	1.01	0.00	0.00	0.47	0.00	0.3 ± 0.4
nerolidol	1565	1, 2	2.13	3.04	3.17	2.50	2.71	3.32	2.81 ± 0.45
epi-globulol	1573	1, 2	0.00	0.00	0.00	0.00	0.00	0.48	0.08 ± 0.2
palustrol	1583	1, 2	0.00	0.00	0.00	3.78	0.00	4.11	1.32 ± 2.04
spathulenol	1590	1, 2	2.40	0.00	0.56	0.98	0.98	0.57	0.91 ± 0.81
caryophyllene oxide	1597	1, 2	0.00	0.00	0.00	0.00	0.00	0.00	nd
globulol	1598	1, 2	0.00	0.00	0.00	0.00	0.00	0.00	nd
viridiflorol	1607	1, 2	**39.05**	**24.12**	**15.24**	**25.02**	**23.44**	**20.57**	**24.57 ± 7.93**
γ-eudesmol	1647	1, 2	0.00	0.00	0.00	0.00	0.00	0.00	nd
1(5),6-guaiadiene	1662	1, 2	0.00	0.00	0.00	0.00	0.00	0.00	nd
β-eudesmol	1665	1, 2	0.00	0.00	0.00	0.51	0.00	0.00	0.09 ± 0.21
α-eudesmol	1669	1, 2	0.00	0.80	0.15	0.00	1.98	2.64	0.93 ± 1.13
τ-muurolol	1666	1, 2	0.00	0.00	0.00	0.00	0.00	0.00	nd
α-bisabolol	1692	1, 2	0.00	0.00	0.00	0.00	0.04	0.00	t
tetracosane	2300	1, 2	0.00	0.00	0.00	0.00	0.00	0.00	nd
heneicosane	2264	1, 2	0.00	0.00	0.00	0.00	0.00	0.00	nd
% Identification			91.89	79.75	88.96	81.81	87.74	80.45	85.1 ± 5.08
**Grouped components:**									
Monoterpene hydrocarbons			34.56	25.19	35.67	28.56	33.21	25.76	30.49 ± 4.58
Oxygenated monoterpenes			5.61	4.01	28.13	8.93	8.95	5.67	10.22 ± 9
Sesquiterpene hydrocarbons			7.79	22.59	5.97	11.52	16.47	17.32	13.61 ± 6.31
Oxygenated sesquiterpenes			43.93	27.96	19.18	32.80	29.10	31.70	30.78 ± 8.04
Hydrocarbons			0.00	0.00	0.00	0.00	0.00	0.00	nd

RRI: Relative retention indices calculated against n-alkanes, ID: identification method—1 = comparison of retention index; 2 = comparison of mass spectra with MS libraries, nd: not detected, t: trace, major compounds (% > 5) highlighted in **bold**.

The variability and potential diagnostic value of the essential oil data was investigated, and a principal component analysis (PCA) indicates three groupings for the three species from the first (14.70%) and second (10.40%) principal components (Figure 7). The *S. dentata* group shows almost no overlap with the *S. africana* and *S. chamelaeagnea* groups. 

The *Salvia dentata* plots grouped to the left of the scatterplot due to the higher content of camphene (μ = 10.18 ± 2.32; *p* < 0.05), camphor (μ = 14.37 ± 4.43; *p* > 0.05), α-pinene (μ = 11.43 ± 2.42; *p Salvia chamelaeagnea* samples clustered together due to viridiflorol (μ = 24.57 ± 7.93; *p* < 0.05) and limonene (μ = 10.58 ± 9.36; *p* < 0.05). *The S. africana* group was determined by *para*-cymene (μ = 1.30 ± 1.23; *p* > 0.05), caryophyllene oxide (μ = 1.79 ± 1.68; *p* > 0.05) and clovene (μ = t; *p* > 0.05).

The EO chemistry of *S. africana* and *S. chamelaeagnea* are highly similar due to shared major and minor compounds. This is evident from the higher correlation seen between the EOs of *S. africana* and *S. chamelageanea* (S_corr_ > 0.60) as compared with the two against *S. dentata* EOs (S_corr_ > 0.35) (Figure 8). This is likely due to *S. africana* and *S. chamelaeagnea* EOs containing higher amounts of viridiflorol, limonene and β-caryophyllene whereas *S. dentata* EOs contain high levels of camphene, camphor, 1,8-cineole, bornyl acetate and α-pinene. The chemical profile of *S. dentata* is very similar to that of *S. officinalis* reported from the North of Tunisia in containing notable levels of camphor (33.61%), 1,8-cineole (22.22%), α-thujone (21.43%) and camphene (4.88%) [39].

In another study by Tundis et al. [35], the authors analyzed Mediterranean samples of *S. officinalis* from several localities and included data from previous studies in their analyses. Their data also observed high levels of camphor (16.16–18.92%), camphene (6.27–8.08%), 1,8-cineole (eucalyptol) (8.80–9.86%), β-pinene (3.08–9.14%) and α-thujone (1.17–9.26%). 

Other *Salvia* species reported with high levels of camphor and 1,8-cineole include *S. aytachii* Vural & Adigüzel [40] from Turkey and *S. fruticosa* from Lybia [41]. Alone, each compound has been reported to exhibit potent biological activity. For example, camphor, a natural product derived from the wood of *Cinnamomum camphora* L., though also present in other aromatic plant species, such as *Ocimum kilimandscharicum* Gürke (Lamiaceae), is a major source of the compound in Asia [42]. Camphor has been reported to be a counterirritant, rubefacient, mild analgesic, components of liniments for the relief of fibrositis, neuralgia and is a mild expectorant. Camphor oil has been used intramuscular or subcutaneously as a circulatory/respiratory stimulant though these methods are considered hazardous [42,43]. Eucalyptol (1,8-cineole) is a mucolytic monoterpene that has been shown to exhibit anti-inflammatory, bronchodilatory and antimicrobial activity and has been used effectively in the treatment of asthma and associated respiratory ailments [44].

The higher levels of both camphor and 1,8-cineole in *S. dentata* may suggest to its use as a traditional remedy in the treatment of microbial infections (i.e., respiratory, gastrointestinal and topical) due to the synergistic effect of these two compounds as reported by Viljoen et al. [45].

## 3. Materials and Methods

### 3.1. Ethnobotanical Data 

Literature searches were conducted by searching several scientific electronic databases, including GoogleScholar (www.scholar.google.com), EBSCOhost (www.ebsco.com), PubMed (www.pubmed.ncbi.nlm.nih.gov), ScienceDirect (www.sciencedirect.com), SciFinder (www.scifinder.cas.org), Springer (www.springer.com) and Wiley Online Library (www.onlinelibrary.wiley.com). Key words were used to search for literature, and this was conducted in the following manner: (“Species name” AND “synonyms” AND “med*”) and (“Species name” AND “synonyms” AND “traditional use”). A collection of scientific papers, books, dissertations and theses, and unpublished sources were also compiled. 

### 3.2. Plant Material and Isolation of Essential Oil

Fresh aerial parts of *Salvia africana* (n = 6), *S. chamelaeagnea* (n = 6), and *S. dentata* (n = 13), were collected from natural populations in the Cape region of South Africa (Table 5). The three species were identified by B.-E. Van Wyk and voucher specimens are housed at the University of Johannesburg Herbarium (JRAU), University of Johannesburg, South Africa. 

The plant material was airdried at room temperature (indoors, at ca. 18 to 22 °C) for several days until no moisture was present. The essential oils were isolated by subjecting the dried leaf material to hydrodistillation using a Clevenger apparatus for three hours. The resulting EO was stored in amber vials at +4 °C until tested and all results were expressed based on dry matter weight.

### 3.3. Chemical Analyses

*Gas chromatography and gas chromatography coupled to mass spectrometry*—A modified method based on Boukhatem, Kameli and Saidi [46] was used where EO samples were prepared by diluting 20 μL EO in 1000 μL acetone. One microliter of the diluted sample was injected onto a Thermo Scientific™ TRACE™ 1300 gas chromatograph (Rodano, Milan-Italy) equipped with a ZB-5Ms capillary column (30 m × 0.25 mm, 0.25 μm), coupled with a Thermo Scientific™ TSQ™ 8000 Triple Quadrupole mass spectrometer (Austin, TX, USA). The GC-MS system was coupled to a Triplus RSH Analytics PAL autosampler (Switzerland). Chromatographic separation of the essential oil was performed on a non-polar ZB-5Ms (30 m, 0.25 mm ID, 0.25 µm film thickness) Zebron capillary column. Helium was used as the carrier gas at a flow rate of 1 mL/min. The injector temperature was maintained at 250 °C. A total of 1µl of the sample was injected in 50:1 split mode. The oven temperature was programmed as follows: initial temperature of 40 °C held for 5 min; and finally ramped up to 250 °C at a rate of 8 °C/min and held for 5 min with a total runtime of 36 min. The MSD was operated in a full scan mode and the source and quad temperatures were maintained at 230 °C and 150 °C, respectively. The transfer line temperature was maintained at 250 °C. The mass spectrometer was operated under electron impact mode at ionization energy of 70 eV, scanning from 35 to 500 m/z. The *Salvia* EOs constituents were tentatively identified by comparing their Retention Indices (RI) with those from literature such as Adams [47] and the similarity of their mass spectra with those databased in the NIST-MS library. RI were calculated under the same operating conditions in relation to a homologous series of *n*-alkanes (C8–C24).

### 3.4. Statistical Analyses 

Principle component analysis was done using Rstudio (Rstudio Team (2021). Rstudio: Integrated Development Environment for R. Rstudio, PBC, Boston, MA, USA. http://www.rstudio.com/). The correlation analysis by UPGMA was done using PAST 4.1 software (Copyright Hammer & Harper; free download from: https://www.nhm.uio.no/english/research/infrastructure/past/).

## 4. Conclusions

Although the three species of blue-flowered sages are closely related, *Salvia dentata* presents a distinct chemical profile from the other two species (*S. africana* and *S. chamelaeagnea*) due to the higher quantities of camphene, camphor, bornyl acetate and 1,8-cineole (eucalyptol) present as highlighted by the data. For the first time, the essential oil chemistry has been reported for this medicinally relevant species and presents opportunities for further investigation within the fields of biological activity and phenolic chemistry.

## Figures and Tables

**Figure 1 molecules-27-06478-f001:**
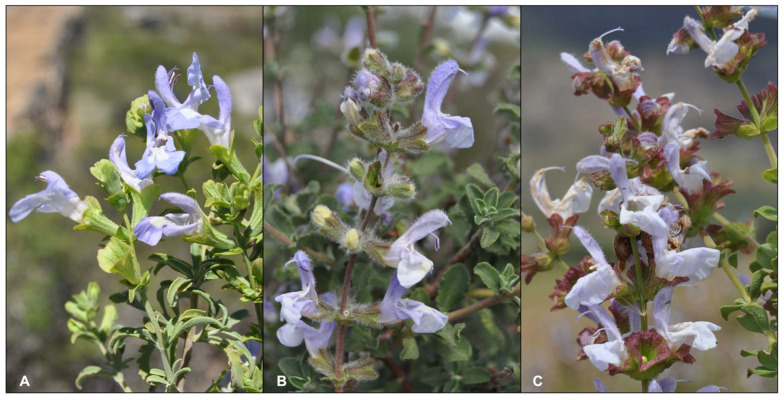
Three closely related, blue-flowered South African medicinal *Salvia* species. (**A**) *Salvia dentata*, (**B**) *S. africana* and (**C**) *S. chamelaeagnea.* All photographs taken by B.-E. Van Wyk.

**Figure 2 molecules-27-06478-f002:**
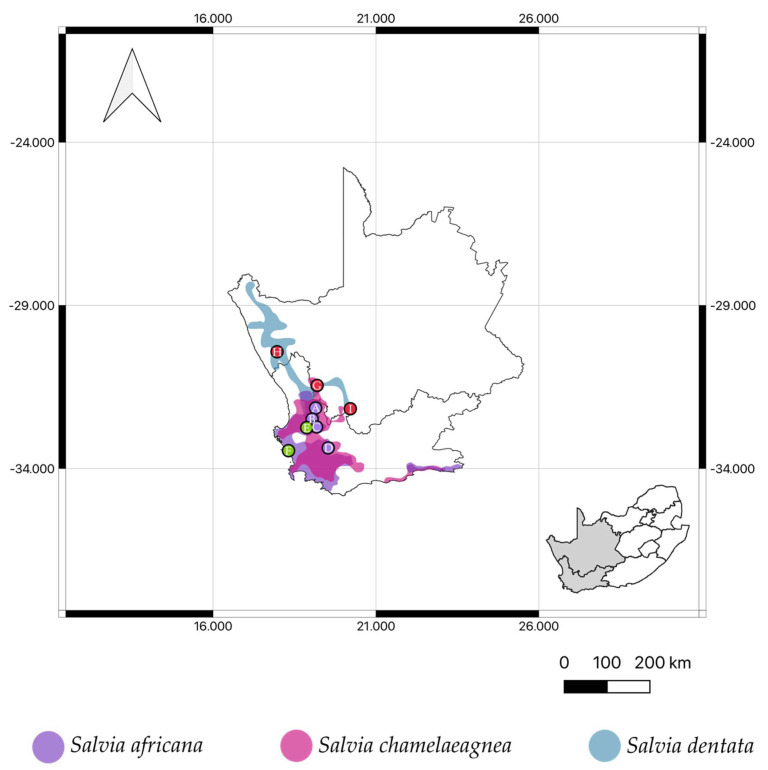
Distribution of the three medicinal, blue-flowered *Salvia* species in the Western and Northern Cape provinces of South Africa. Localities where plant materials were collected for analysis are numbered as in the material and methods section (data points A–I).

**Figure 3 molecules-27-06478-f003:**
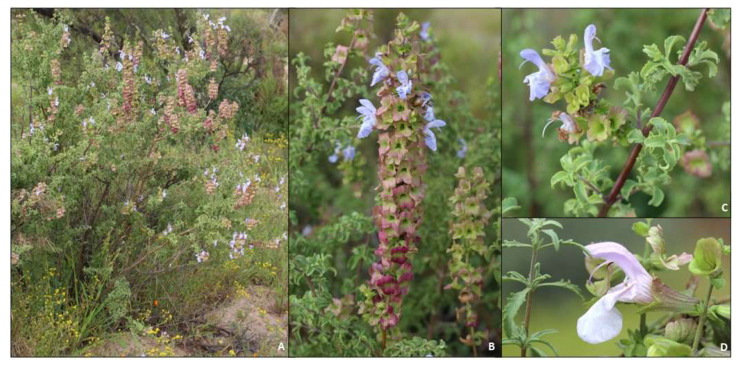
Morphology and diagnostic characters of *Salvia dentata*. (**A**) habit (near Garies); (**B**) inflorescence (near Garies); (**C**) leaves and flowers (Nieuwoudtville; note dentate leaf margins); (**D**) leaves and flower (near Middelpos; note dentate leaves and glandular calyx). All photographs taken by B.-E. Van Wyk.

**Figure 4 molecules-27-06478-f004:**
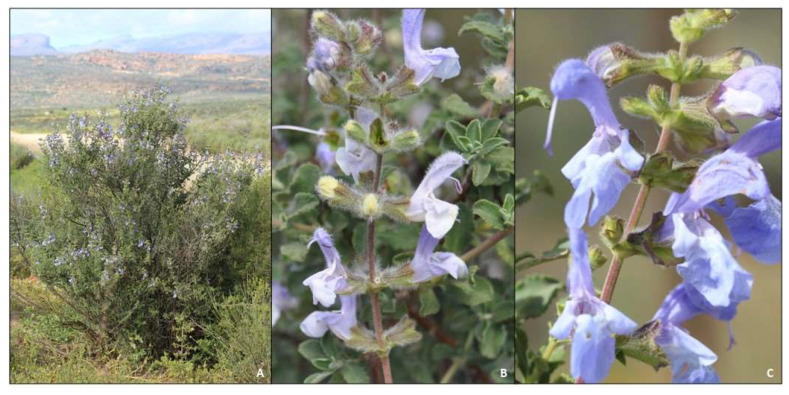
Morphology and diagnostic characters of *Salvia africana*. (**A**) habit (Cederberg); (**B**) inflorescence and leaves (Malmesbury); (**C**) flowers (Cederberg; note hirsute calyx). All photographs taken by B.-E. Van Wyk.

**Figure 5 molecules-27-06478-f005:**
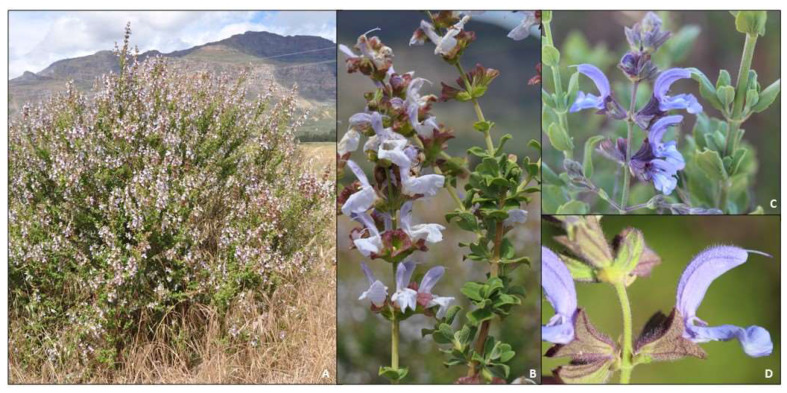
Morphology and diagnostic characters of *Salvia chamelaeagnea*. (**A**) habit (Piketberg); (**B**) inflorescence and leaves (Piketberg); (**C**) leaves and flowers (Darling); (**D**) flower (note glandular calyx). All photographs taken by B.-E. Van Wyk.

**Figure 6 molecules-27-06478-f006:**
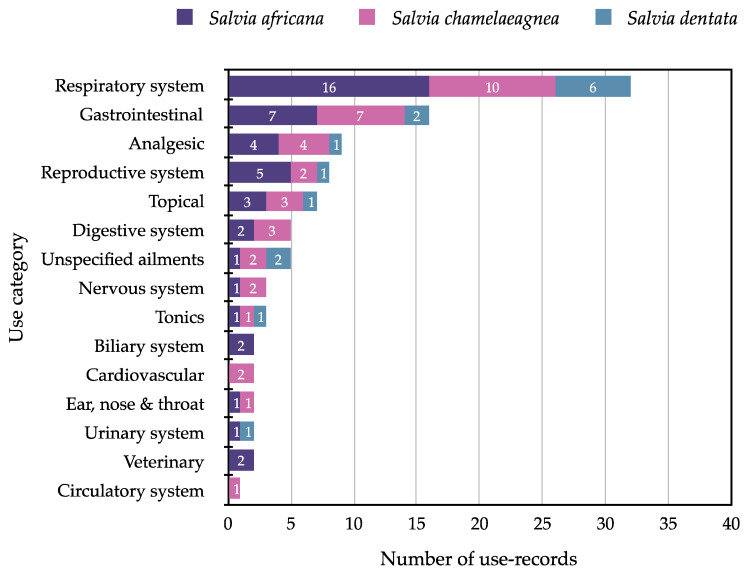
Number of medicinal use-records per use category for the three Cape blue-flowered *Salvia* species.

**Figure 7 molecules-27-06478-f007:**
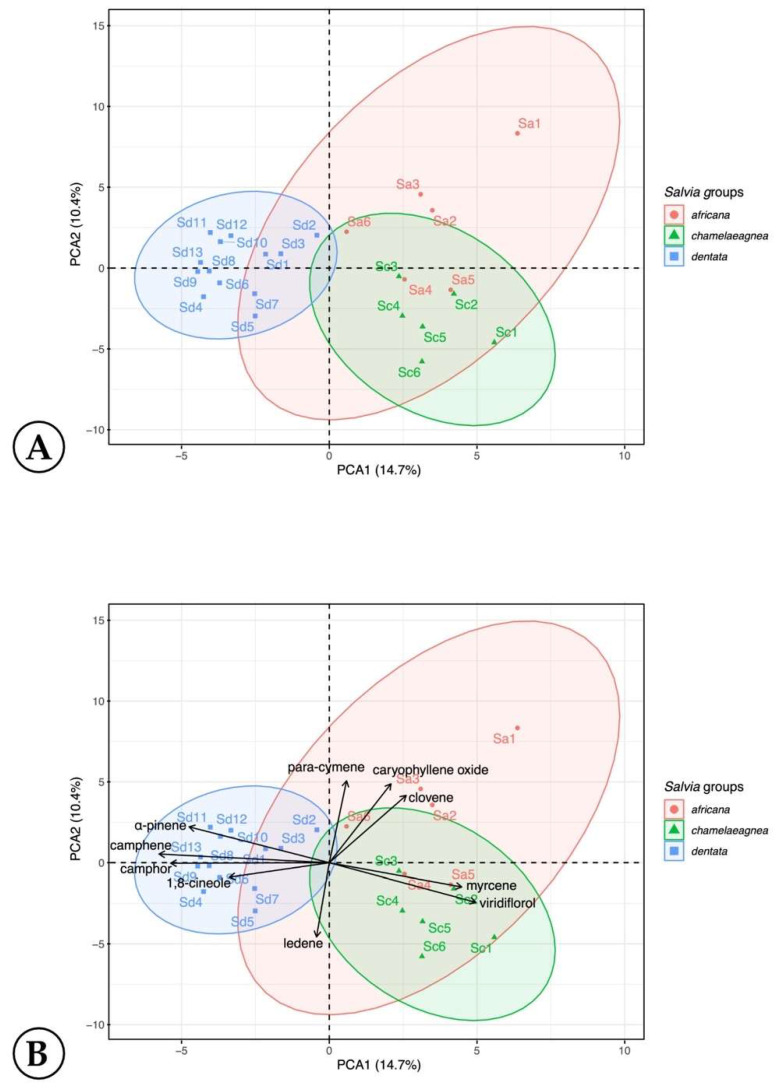
Principal component analysis (PCA) scatterplot (biplot) of the first (14.70%) and second (10.40%) components; (**A**) scatterplot of species and (**B**) scatterplot of the essential oils (loadings plot).

**Figure 8 molecules-27-06478-f008:**
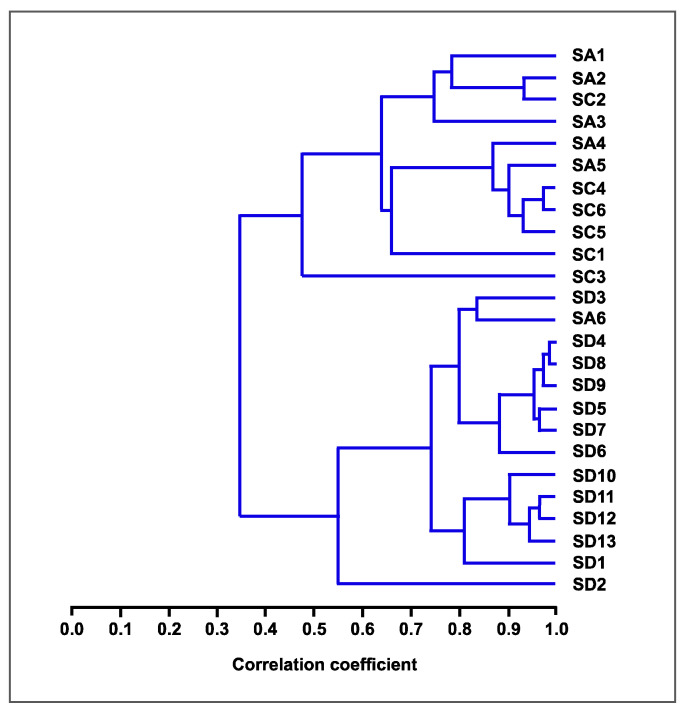
A dendrogram generated by cluster analysis of the percentage composition of essential oils from the three *Salvia* species studied, based on correlation and using the unweighted pair-group method with arithmetic average (UPGMA). SA = *S. africana*, SC = *S. chamelaeagnea* and SD = *S. dentata*.

**Table 5 molecules-27-06478-t005:** Sample information.

Species	Voucher Numbers	Locality	Point on Map (Figure 2)	Dry Weight (g)
*Salvia africana*	RDR and BEVW 0048A	Traveller’s Rest, Cederberg, WC	A	33.46
	RDR and BEVW 0048B	Traveller’s Rest, Cederberg, WC	A	45.1
	RDR and BEVW 0048C	Traveller’s Rest, Cederberg, WC	A	60.08
	RDR and BEVW 0076	Citrusdal, WC	B	57.1
	RDR and BEVW 0077	Elandskloof, WC	C	61.27
	RDR and BEVW 0078	Theronsberg Pass, WC	D	101.38
*Salvia chamelaeagnea*	RDR and BEVW 0055A	Piketberg, WC	E	59.97
	RDR and BEVW 0055B	Piketberg, WC	E	39.91
	RDR and BEVW 0055C	Piketberg, WC	E	46.37
	RDR and BEVW 0073A	Darling, WC	F	39.64
	RDR and BEVW 0073B	Darling, WC	F	41.45
	RDR and BEVW 0073C	Darling, WC	F	30.17
*Salvia dentata*	RDR and BEVW 0046A	Nieuwoudtville, NC	G	89.2
	RDR and BEVW 0046B	Nieuwoudtville, NC	G	71.4
	RDR and BEVW 0046C	Nieuwoudtville, NC	G	73.2
*Salvia dentata* Cont.	RDR and BEVW 0047A	Spoegrivier road near Garies, NC	H	73.43
	RDR and BEVW 0047B	Spoegrivier road near Garies, NC	H	124.45
	RDR and BEVW 0047C	Spoegrivier road near Garies, NC	H	149.8
	RDR and BEVW 0047D	Spoegrivier road near Garies, NC	H	66.49
	RDR and BEVW 0047E	Spoegrivier road near Garies, NC	H	69.31
	RDR and BEVW 0047F	Spoegrivier road near Garies, NC	H	151.35
	RDR and BEVW 0081A	Tankwa, NC	I	86.76
	RDR and BEVW 0081B	Tankwa, NC	I	53.38
	RDR and BEVW 0081C	Tankwa, NC	I	84.51
	RDR and BEVW 0081D	Tankwa, NC	I	75.14

WC = Western Cape and NC = Northern Cape.

## Data Availability

The geographical data presented in this study are openly available in GBIF (Global Biodiversity Information Facility) at https://doi.org/10.15468/dl.sa4wq9.

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
