# Peer review of "Essential Oil Composition and Traditional Uses of Salvia dentata, a Poorly Known Medicinal Plant from Namaqualand, South Africa"

_molecules, 2022, doi:10.3390/molecules27196478_

Round 1
Reviewer 1 Report
In: Title
- the title should be changed.
In: Abstract
- the sentence “Camphor and 1,8-cineole have been previously…” should be excluded.
- some corrections of the text are necessary, because it is not clear enough now.
In: Keywords
- the keywords should be more specific.
In: Introduction
- the aim of the paper is not clear.
In: Results and Discussion
- the information on the application of the plants described in table 1 and figure 5 should be given in part Introduction.
- information about the color and odor of the essential oils is missing.
- it should be given the explanation about the difference of the concentration of volatile compounds.
- what are the allergens in the essential oils?
- the explanation about different chemical composition of the samples is missing.
In: Tables 2, 3 and 4
- “humulene” should be written instead of “humelene”.
- “muurolol” should be written instead of “muurlol”.
In: Material and methods
- information on plant samples is missing.
- the moisture content (%) of the dried plant, the method for its determination and the conditions of the drying should be specified. The essential oil yield should be calculated in % to absolute dry weight.
In: Conclusions
- some corrections of the text are necessary, because it is not clear enough now.
- the references should be excluded from the text.
Author Response
Detailed response to reviewer 1
We thank the reviewer for the time spent on trying to improve our paper.
Comments and Suggestions for Authors
In: Title
- the title should be changed.
In: Abstract
- the sentence “Camphor and 1,8-cineole have been previously…” should be excluded.
** This has now been excluded
- some corrections of the text are necessary, because it is not clear enough now.
** We tried to improve the text, hopefully it is acceptable now.
In: Keywords
- the keywords should be more specific.
** More specific keywords have been included
In: Introduction
- the aim of the paper is not clear.
** We tried to improve the text, hopefully it is now much better.
In: Results and Discussion
- the information on the application of the plants described in table 1 and figure 5 should be given in part Introduction.
** This change has been made – thank you
- information about the color and odor of the essential oils is missing.
** This information has now been added – thank you for the suggestion
- it should be given the explanation about the difference of the concentration of volatile compounds.
** This information has now been added – thank you for the suggestion
- what are the allergens in the essential oils?
** This is beyond the scope of the study and was thus not included
- the explanation about different chemical composition of the samples is missing.
** This information has now been added – thank you for the suggestion
In: Tables 2, 3 and 4
- “humulene” should be written instead of “humelene”.
** The correction has been made – thank you drawing our attention to the error
- “muurolol” should be written instead of “muurlol”.
** The correction has been made – thank you drawing our attention to the error
In: Material and methods
- information on plant samples is missing.
** This information has now been added – thank you for the suggestion
- the moisture content (%) of the dried plant, the method for its determination and the conditions of the drying should be specified. The essential oil yield should be calculated in % to absolute dry weight.
** This information has now been added – thank you for the suggestion
In: Conclusions
- some corrections of the text are necessary, because it is not clear enough now.
** Some of the text has been rearranged and should now read better
- the references should be excluded from the text.
** The references have now been excluded
Submission Date
19 August 2022
Date of this review
24 Aug 2022 07:12:18
Reviewer 2 Report
This manuscript (molecules-1898725) investigated the oil composition in Salvia dentata, a medicinal plant from South Africa, and provided a comparison with the other two closely related Cape bloublomsalies (Salvia africana and S. chamelaeagnea). This research is of significance for understanding the medical value of medicinal plant and discovering this potential application. The manuscript should be improved as following:
1. I suggest that morphology characters of three plant species could be moved into the section of Results, and should be described with more details.
2. The manuscript should cite the references the experimental methods referred in the section of ‘Materials and Methods’.
3. Format the manuscript according to the guideline for authors.
Author Response
Detailed response to reviewer 2
We thank the reviewer for the time spent on trying to improve our paper.
Comments and Suggestions for Authors
This manuscript (molecules-1898725) investigated the oil composition in Salvia dentata, a medicinal plant from South Africa, and provided a comparison with the other two closely related Cape bloublomsalies (Salvia africana and S. chamelaeagnea). This research is of significance for understanding the medical value of medicinal plant and discovering this potential application. The manuscript should be improved as following:
- I suggest that morphology characters of three plant species could be moved into the section of Results, and should be described with more details.
** This change has been made and we thank the reviewer of the suggestion – the addition of the morphological characters to material and methods adds more detail to the species of interest
- The manuscript should cite the references the experimental methods referred in the section of ‘Materials and Methods’.
** Relevant references have now been added and we thank the reviewer of the suggestion
- Format the manuscript according to the guideline for authors.
** This change has been made and the manuscript follows the guidelines set out by the publishers
Submission Date
19 August 2022
Date of this review
27 Aug 2022 11:02:06
Reviewer 3 Report
The manuscript is focused on the analysis of the essential oil of a lesser-known African plant. The text is written comprehensibly and quite clearly. I am very positive about the successful image attachment documenting the appearance of important Salvia species. This is a pleasant addition to the overview of the issue from a botanical point of view.
I have the following comments and suggestions for the manuscript:
1/ Chapter 4.2 - I lack more detailed information on the isolation of EO (the current text contains only one sentence L. 234-235).
2/ Fig 5 - please edit the font in the description of the horizontal axis ("No.") - apparently a different font was used compared to other parts of the object.
3/ Especially in chapter 2.2, the citation in the text needs to be corrected - e.g. L. 118 - it seems to me that the citation does not look very appropriate in this form (e.g. "Fisher (2005) (11)"... etc.), even on L. 120 there is a completely inappropriately written quotation with the insertion of the result in %..
4/ Table 2, 3, etc. - I suggest changing the name of the object to "Identified essential oil components (%)..."
Author Response
Detailed response to reviewer 3
We thank the reviewer for the time spent on trying to improve our paper.
Comments and Suggestions for Authors
The manuscript is focused on the analysis of the essential oil of a lesser-known African plant. The text is written comprehensibly and quite clearly. I am very positive about the successful image attachment documenting the appearance of important Salvia species. This is a pleasant addition to the overview of the issue from a botanical point of view.
I have the following comments and suggestions for the manuscript:
1/ Chapter 4.2 - I lack more detailed information on the isolation of EO (the current text contains only one sentence L. 234-235).
** More detail regarding the essential oil isolation has been added – we thank the reviewer for the suggestion
2/ Fig 5 - please edit the font in the description of the horizontal axis ("No.") - apparently a different font was used compared to other parts of the object.
** This change has been made - we thank the reviewer for drawing our attention to that detail
3/ Especially in chapter 2.2, the citation in the text needs to be corrected - e.g. L. 118 - it seems to me that the citation does not look very appropriate in this form (e.g. "Fisher (2005) (11)"... etc.), even on L. 120 there is a completely inappropriately written quotation with the insertion of the result in %..
** Citations have now been corrected for this section
4/ Table 2, 3, etc. - I suggest changing the name of the object to "Identified essential oil components (%)..."
** This change has been made – we thank the reviewer for this suggestion
Submission Date
19 August 2022
Date of this review
02 Sep 2022 22:24:41
Round 2
Reviewer 1 Report
In: Title
- I think that the title should be changed, because it does not match the content of the article.
In: Results and Discussion
- I think that the information on the application of the plants described in Table 1 and in Figure 6 should be given in part Introduction.
In: Table 2, 3 and 4
- "oxygenated sesquiterpenes" should be written instead of "oxygenated sesquiterpene".
In: Material and methods
- line 313: it should be specified what “room temperature” is.
- "min" should be written instead of "minutes".
- "mL" should be written instead of "ml".
Author Response
Comments and Suggestions for Authors In: Title - I think that the title should be changed, because it does not match the content of the article. ###It is a pity that no indication is given of what is missing, or how the title should be changed. In our opinion, the tile exactly matches the content, namely a study of the essential oil composition of Salvia dentata (reported for the first time) and the traditional medicinal uses of Salvia dentata (reviewed and reported for the first time, from our own field work results). The comparisons with the other two Salvia species are secondary, as these species have been studied before. We simply included them in the paper to have good comparative data (i.e., essential oil results obtained in the same laboratory using the same methods). In: Results and Discussion - I think that the information on the application of the plants described in Table 1 and in Figure 6 should be given in part Introduction. ###We respectfully disagree. Perhaps the reviewer is not aware that the details given in Table 1 and in Figure 6 are new results, generated as a detailed review from our own research, and not only taken directly from the literature, but also from unpublished dissertations and theses from our research group. This review is part of the main findings of the paper and therefore, in our considered opinion, fit best in the results section. In: Table 2, 3 and 4 - "oxygenated sesquiterpenes" should be written instead of "oxygenated sesquiterpene". ###Yes indeed, thank you for pointing out this inconsistency (now corrected). In: Material and methods - line 313: it should be specified what “room temperature” is. ###Yes, details now added, thank you. - "min" should be written instead of "minutes". ###Yes, done, thank you. - "mL" should be written instead of "ml". ###Yes, done, thank you.